# Variation in the Floral Morphology of *Prosthechea karwinskii* (Orchidaceae), a Mexican Endemic Orchid at Risk

**DOI:** 10.3390/plants13141984

**Published:** 2024-07-20

**Authors:** María Hipólita Santos-Escamilla, Gabriela Cruz-Lustre, Manuel Cuéllar-Martínez, Luicita Lagunez-Rivera, Rodolfo Solano

**Affiliations:** 1Instituto Tecnológico del Valle de Oaxaca, Ex Hacienda Nazareno, Santa Cruz Xoxocotlán 71233, Oaxaca, Mexico; mhsantosescamilla55@gmail.com; 2Departamento de Botânica, Instituto de Ciências Biológicas, Universidade Federal de Minas Gerais, Avenida Antônio Carlos 6627, Pampulha, Caixa Postal 486, Belo Horizonte 31270-910, Minas Gerais, Brazil; 3Centro Interdisciplinario de Investigación para el Desarrollo Integral Regional Unidad Oaxaca, Instituto Politécnico Nacional, Santa Cruz Xoxocotlán 71230, Oaxaca, Mexico; manuel.cuellarm@gmail.com; 4Laboratorio de Extracción y Análisis de Productos Naturales Vegetales, Centro Interdisciplinario de Investigación para el Desarrollo Integral Regional Unidad Oaxaca, Instituto Politécnico Nacional, Santa Cruz Xoxocotlán 71230, Oaxaca, Mexico; llagunez@ipn.mx

**Keywords:** endangered species, epiphyte, floral variation, intraspecific variation, morphometrics, orchids, ornamental/useful species

## Abstract

*Prosthechea karwinskii* is an orchid endemic to Mexico, threatened by the destruction of its habitat and the extraction of specimens to meet its demand for ornamental and religious use. Most of its populations, including the most locally abundant ones, are found in Oaxaca state. Variations in some floral traits have been observed in these populations. We implemented a morphometric analysis to assess their floral variation and identify the most significant characters in the morphological patterns of this orchid. Floral samples were collected from 17 populations of *P. karwinskii* in Oaxaca, as well as from specimens used as ornaments during Easter in an Oaxacan community (Zaachila), whose origin is unknown. Sampling of natural populations covered the environmental, geographic, and morphological variation of the species. We performed an analysis of variance (ANOVA), principal component analysis (PCA), canonical variate analysis (CVA), and cluster analysis, including 185 individuals and 45 variables (12 of them were discarded in the multivariate analyses due to high correlation). Characters of the column, lateral sepal, and labellum were most informative for the observed morphological patterns. Albarradas showed the greatest morphological differentiation, mainly due to the column. In general, individuals from the same locality tended to overlap more, especially the populations of Jaltianguis and Yahuiche, which were different from the geographically close population of Etla. Teposcolula presented the highest values in perianth characters, unlike Sola_Rancho Viejo. The specimens recovered from religious ornaments were morphologically more similar to those from Yanhuitlan and Etla. This morphometric analysis identified characters as potential taxonomic markers for *P. karwinskii* and related species, showing its potential to associate specimens of unknown origin with their probable geographical region. Our work encourages working on collaborative conservation strategies to ensure the long-term permanence of both the species and its traditional uses.

## 1. Introduction

*Prosthechea karwinskii* (Mart.) J.M.H. Shaw is an orchid endemic to western and southern Mexico, inhabiting mountainous regions where it grows as a hanging epiphyte in oak or oak-pine forests subject to a well-marked seasonal drought (Figure 1A). The inflorescence is a two-flowered (exceptionally three flowers) raceme arising from the last developed pseudobulb (Figure 1B). The taxon was described in 1830, but its taxonomic history has been linked to that of *Prosthechea citrina* (Lex.) W.E. Higgins, or any of its nomenclatural synonyms, a very similar species, with which it becomes sympatric in some locations in Guerrero and Michoacán. In the past, the information available for *P. karwinskii* was invariably attributed to *P. citrina*; only in recent years have both species come to be considered different [1,2]. This species is one of the most distinctive orchids in the Mexican flora, valued for the ornamental beauty and pleasant aroma of its flowers, as well as having cultural significance since pre-Hispanic times. It was one of the orchids from which mucilage was obtained and used as an adhesive in feather art in pre-Hispanic times and during the colonial period [3]. In traditional medicine, it has been used to soothe coughs, heal wounds and burns, treat diabetes, prevent miscarriage, and aid in childbirth [4,5,6,7]. The flowers are used as decorations in homes, commercial stands, and temples during Easter commemorations [8,9]. Additionally, due to the beauty, color, and aroma of its flowers, this plant is cultivated in a rustic manner in orchards in many communities in Oaxaca.

In most of the localities where *P. karwinskii* grows, it is locally scarce and survives in forest fragments surrounded by a matrix of environments modified by anthropogenic causes (conversion into crop fields, human settlements, opening of new roads, goat farming). In Oaxaca, the most abundant populations of the species occur, but they face the risk of extraction for temporary adornment in local trade [8,9,10], for religious purposes [8], and to a lesser extent, for medicinal use [4]. This practice occurs annually, mainly during the species’ flowering season, leading to its inclusion in the list of species of Mexican wild flora at risk [11]. For other epiphytic orchids growing in the mountain forests of Mexico, it has been demonstrated that extraction for local trade has effects on species subjected to this practice, such as reductions in population size and rates of fertility and recruitment, as well as loss of genetic diversity [12,13,14,15].

In the localities of *P. karwinskii* in Oaxaca, variation has been observed in some floral traits, such as flower size and coloration, the shape of the labellum, and the shape of the apical teeth of the column. This leads to the assumption of the existence of intraspecific variation, at least among populations in the state, which has not been analyzed either through the use of morphological or molecular markers. The analysis of this variation will be important for identifying morphotypes with ornamental potential and desirable in a management program, recognizing phytogenetic diversity present in the species, identifying forms or subspecies within the orchid, as well as determining a possible geographical pattern associated with morphological variation.

Morphometric studies have been employed in some species of Orchidaceae, primarily analyzing variation associated with floral morphology [16,17,18,19,20,21,22], although vegetative morphology has also been considered, including attributes of leaf anatomy [17,22]. Studies in this regard seek to find intraspecific differences [18,23] and interspecific differentiation to delimit similar taxa considered as cryptic species [16,22], recognize taxa of hybrid origin [24], or identify morphotypes with phytogenetic value [19,20]. The quantitative analysis of characters identified in the labellum has been valuable for recognizing and characterizing intraspecific variation in orchids [19,20], although the use of traits present in other floral structures has also been useful [18]. Morphometry has also been used to trace the geographical origin of samples of unknown origin, particularly for species or products of economic importance [25,26,27,28,29,30]. Although various sources of information (e.g., genetic and chemical) and analytical tools can address this issue, the use of multivariate methods with morphological characters offers the advantage of low cost [27,28,29,30] and relatively easy data collection for a large number of individuals [26,27]. However, this topic is analytically complex and requires caution in its implementation and interpretation due to the need for a robust reference and the requirements of the analyses [26,28].

The objective of this study was to analyze the variation among populations from different localities in Oaxaca, Mexico, and to identify variables with taxonomic potential, through a morphometric analysis. Additionally, an attempt was made to determine if this set of characters allows relating specimens extracted from their habitats and whose origin is unknown, which were recovered after having had a religious use. Predictions for this study are as follows: (1) the floral morphology of *P. karwinskii*, analyzed using morphometric methods, will allow us to recognize the interpopulation variation of the species; and (2) the floral traits of *P. karwinskii* could serve as a morphological marker to associate the geographical origin of specimens from unknown localities.

## 2. Results

### 2.1. ANOVA and Kruskal–Wallis Tests

Most of the 40 floral variables analyzed with ANOVA showed significant differences among the localities of *P. karwinski* (Table 1). However, the characters lengths between maximum width and apex (SlLa) and angle at the apex (SlAa) of the lateral sepal, width at 1/3 (SdA1), length between maximum width and apex (SdLa), and angle at the apex (SdAa) of the dorsal sepal, and width at the base of the middle lobe of the labellum (LaAbm) did not show significant differences among localities. The specimens from Teposcolula exhibited the highest values for the length and width of floral segments, which determine flower size, such as total length (SlLt) and maximum width (SlAm) of the lateral sepal, total length (SdLt) and maximum width of the dorsal sepal (SdAm), total length (PeLt) and maximum width (PeAm) of the petal, total length (LaLt) and maximum width (LaAm) of the labellum, and total length of the column (CoLt) (see Appendix A). In contrast, the specimens from Sola_Rancho Viejo appeared to have the smallest flowers, as characters like total length (SdLt) and maximum width (SdAm) of the dorsal sepal, total length (PeLt) and maximum width (PeAm) of the petal, total length (LaLt), and maximum width (LaAm) of the labellum showed the lowest values (see Appendix A). Additionally, the locality of Albarradas showed the highest values in two characters of the labellum (LaAml, LaA1l), in the separation between the teeth of the column (AnDlDm) and in the width of the cuniculus near the base (CuA1). However, it presented the lowest values for one character of the petal (PeAa), two in the labellum (LaAul, LaAlm) and three in the teeth of the column (DlAl, DlAn, DmAl).

The Kruskal–Wallis test revealed significant differences among localities (df = 11, *p*-value < 0.05) for the thickness at the middle part (CoGm, X^2^ = 39.759) and the anther level of the column (CoGa, X^2^ = 40.521), and width of its middle tooth (DmAn, X^2^ = 27.5236). However, there were no differences among localities (df = 11, *p*-value > 0.05) for the apex angle of the middle lobe of the labellum (LaAam, X^2^ = 18.374) and width of the column at the stigma level (CoAe, X^2^ = 17.823). The specimens from Teposcolula and Sola_El Lazo recorded the highest values for the thickness of the column at the anther level (CoGa), while the individuals from Zaachila and Albarradas had the lowest values for this character (Appendix A). For thickness at the middle part of the column (CoGm), the individuals from Teposcolula had the highest value. For the width of the middle tooth of the column (DmAn), the individuals from Sola_El Lazo had the highest value, while those from Albarradas had the lowest value.

### 2.2. Multivariate Analyses (PCA, CVA, and Cluster Analysis)

Both the PCA and CVA produced similar results whether the samples obtained in Zaachila were excluded or included. Therefore, for both methods, only the results of the analyses including the floral samples obtained from this community in Oaxaca are presented. For the final analyses (PCA and CVA), 185 individuals and 33 morphological variables of *P. karwinskii* were included.

The PCA showed that eight principal components had eigenvalues > 1.0, which together accounted for 71.91% of the total variance (Table 2). Among the eight components that retained the highest percentage of variance, the first one explained a third of it (33.51%) and was more correlated with variables related to floral size. Specifically, the maximum width of the labellum (LaAm), the petal (PeAm), and the lateral sepal (SlAm) were the most important variables in the morphological patterns observed in this axis. This analysis showed a high overlap among individuals from different populations; however, some individuals from Teposcolula and the populations of Yanhuitlan, Etla, and Sola_Rancho Viejo appeared at the extremes (Figure 2).

In the CVA, three canonical axes had eigenvalues > 1.0 and together explained 66.46% of the total variance (Table 2). The first canonical axis explained 33.56% of the variance and reflected a greater contribution of the height of the middle tooth of the column (DmAl), the maximum width of both the lateral sepal (SlAm) and the lateral lobe of the labellum (LaAml). On this axis, there was a clear separation of individuals from the Albarradas population from the other populations (Figure 3A). The samples from Jaltianguis and Yahuiche showed less dispersion along the first three canonical axes and overlapped with each other. Along axis 1, individuals from these two populations were completely separated from Etla and Zaachila (except for one individual from Jaltianguis). Axis 1 also showed a complete separation of the samples from Zaachila from those from Juquila and the populations of Sola (El Lazo and Lachixío), with only a marginal overlap with the remaining Sola population (Rancho Viejo) and Tlaxiaco. Except for one individual, Teposcolula was also almost completely separated from Zaachila and Etla.

The second axis explained 19.96% of the variance and showed a greater contribution from the total length of both the labellum (LaLt) and the lateral sepal (SlLt), as well as the height of the lateral tooth of the column (DlAl). On axis 2, individuals from the Yahuiche and Jaltianguis populations were completely separated from Tlaxiaco and Teposcolula (Figure 3A) and present a marginal overlap with Juquila and Sola_El Lazo. On this axis, Yanhuitlan is completely separate from Teposcolula and partially overlaps with Sola_Rancho Viejo. The third axis accounted for 12.94% of the variance and had as the most important variables the angle between the lateral lobe and the claw of the labellum (LaAul), the thickness of the column at the level of the anther (CoGa) and the maximum width of the labellum (LaAm). On this axis, the separation of individuals from Albarradas from all other populations was again highlighted (Figure 3B), as seen on axis 1. Along the first three axes, the samples from Zaachila showed greater overlap with those from the populations of Yanhuitlan and Etla (Figure 3).

The cluster analysis also revealed the high morphological divergence of Albarradas population (Figure 4). In the UPGMA dendrogram, this population was externally linked to the group containing the remaining populations. This latter group is divided into three subgroups: the first formed by intermixed populations from the southern regions of Oaxaca (Sola_El Lazo and Juquila) and two populations from the Mixteca (Teposcolula and Tlaxiaco); the second includes the remaining populations from Sola de Vega (Sola_Rancho Viejo and Sola_Lachixío) nested between those of the Sierra Norte (Jaltianguis and Yahuiche); and the third subgroup comprises Zaachila linked to Yanhuitlan, with Etla also joined to them.

## 3. Discussion

### 3.1. Morphological Variation in Natural Populations of Prosthechea karwinskii

Plants have the ability to modify their phenotype in response to environmental conditions. However, the variation within a species due to the environment is expected to be smaller and more limited in floral characters compared to vegetative ones, since the former are related to reproductive success and must maintain their function [31,32]. The variation in floral morphology can be interpreted as an adaptation to selection by different pollinators [33,34]. Interactions between plant-pollinators and climatic influence can explain the variation in floral traits, suggesting that the variation expressed on them is a product of an adaptive response [35]. However, other processes can produce divergence among geographically separated populations of flowering plants, such as random genetic drift, isolation, indirect selection, and genetic factors [36,37]. Because floral traits are considered phenotypically more stable than vegetative ones, their variation tends to be less within populations, making them valuable for recognizing infraspecific variation [19,23]. Studies evaluating infraspecific morphological variation in orchids have so far been conducted using floral traits and have generally been useful in identifying such variation [16,17,18,19,20,21,22,23,24,25].

Of the floral characters evaluated in the ANOVA, most (38 out of 40) showed significant variation among the populations of *P. karwinskii* analyzed here, as demonstrated by the ANOVA and Kruskal–Wallis tests. Of the other five floral characters that did not show normality and were evaluated with the Kruskal–Wallis test, three of them (corresponding to the column) showed significant differences among populations (thickness in the middle part, thickness at the level of the anther, and width of the middle tooth). These tests identified Teposcolula (Mixteca region) as the population in Oaxaca that includes individuals with the largest flowers (showing the highest values for sepal, petal, and labellum length and width), while those from Sola_Rancho Viejo (Sierra Sur region) include individuals with the smallest flowers. Among orchids of ornamental value, specimens with larger flowers are preferred for cultivation and are selected as mother plants or as parents for artificial hybrids. Since *P. karwinskii* is an orchid appreciated for its ornamental value, the Teposcolula population holds greater importance in horticulture compared to other localities in Oaxaca.

The PCA conducted with the set of floral characters for *P. karwinskii* showed a high overlap between individuals from the 17 sampled locations and those obtained from Zaachila. Thus, it seems that the traits of floral morphology do not allow for detecting differences between populations, or their number is not sufficient to discriminate the infraspecific variation of *P. karwinskii*. Nevertheless, along the axis that accumulated the highest percentage of variance, several individuals from Teposcolula and Yanhuitlan appear at the extremes. The characters associated with this pattern were related to the width of the perianth segments (labellum, petal and lateral sepal). Among all the populations analyzed, Teposcolula has the widest segments, while Yanhuitlan, Etla, and Sola_Rancho Viejo have the narrowest segments, as also revealed by the univariate analyses. Ibáñez [27] suggested that complex patterns revealed through multivariate analyses of morphological data may be associated with the life history of the organisms. The localities of Teposcolula and Yanhuitlan host two of the largest populations of the species in Oaxaca, where the highest percentage of individuals are reproductive. Possibly, these two populations present high levels of genetic variation, which could be related to how their individuals are dispersed in the PCA graph. Other studies that have evaluated interspecific [17,22] or intraspecific variation [23] in orchids have also not found differentiation between populations that are geographically separated when analyzing floral characters with PCA. Such results are common given the method’s assumptions, particularly the lack of a priori categorization into groups that would typically minimize intragroup variance and maximize intergroup variance [38]. However, this multivariate method has been useful for recognizing morphotypes in orchid species represented by wild specimens [20] or cultivated ones [19,25] in Mexico.

The analysis of *P. karwinskii*’s floral morphology with the CVA and cluster analysis were informative for discriminating infraspecific variation. These analyses showed that individuals from Alabarradas are well separated from the rest of the localities. Differences in the teeth of the column discriminate these individuals from those from other localities. Additionally, individuals from Jaltianguis and Yahuiche showed low dispersion and tended to overlap with each other. This morphological pattern is congruent with geography since these two locations are very close to each other in the region known as Sierra Norte of Oaxaca. Interestingly, individuals from Etla, the locality closest to the previous two, tend to show low dispersion among themselves and separate from those of Jaltianguis-Yahuiche. The morphological differentiation that is present with respect to Teposcolula is more expected. The CVA showed that the variables with the greatest contribution are traits of the column (height of the middle and the lateral teeth, and thickness at the level of the anther), lateral sepal (maximum width, total length), labellum (maximum width of the lateral lobe, total length, angle between the lateral lobe and claw, and maximum width). Due to the nature of the characters revealed by the analyses as the most important for the morphological patterns of the species, these traits could be related to the attraction of pollinators, as has been hypothesized in other studies analyzing morphological variation in other plants [22,39]. We recommend paying attention to the most important variables presented in this work, as they can be useful as taxonomic markers at the intraspecific level and possibly at the level of the *Prosthechea citrina* complex, the species group to which *P. karwinskii* belongs.

The population of Albarradas turned out to be the most differentiated among those of *P. karwinskii* from Oaxaca. The individuals from this locality exhibit the lowest height of the middle and lateral teeth of the column, the smaller angle between the lateral lobe and claw of the labellum and the thinnest thickness of the column at the level of the anther. Additionally, they have the greatest maximum width of the lateral lobe of the labellum. This population could be recognized as a variety or geographic form of the species, geographically isolated in the central part of the state of Oaxaca. Moreover, the locality of Albarradas is a priority for the conservation of the species, as it represents a unique morphological variant. Unfortunately, the forest where this form grows in Albarradas hosts one of the least numerous populations of this orchid in Oaxaca. It will be interesting to verify if the morphological differentiation of the Albarradas population is related to genetic differentiation, as has been corroborated for other orchid species using vegetative characters [17].

### 3.2. Morphological Patterns of Zaachila Flowers of Unknown Origin

Among the natural populations of *P. karwinskii* used as reference in this study to compare flowers obtained from Easter celebrations in Zaachila (2017–2019), morphometric analyses revealed greater overlap and morphological similarity between the Zaachila material and populations from Yanhuitlan and Etla. Flowers from Zaachila are distinct from those in Albarradas, Jaltianguis and Yahuiche (Sierra Norte), Teposcolula (Mixteca), Juquila and Sola_El Lazo (Southern mountains of Oaxaca). Conversely, Zaachila specimens exhibit varying degrees of overlap with individuals from the Sierra Sur (Sola_Lachixío and Sola_Rancho Viejo) and Mixteca (Tlaxiaco) regions. While the species is relatively common across various forests in Oaxaca, our inference suggests that individuals rescued from Zaachila could come from the Mixteca or vicinity of Etla. However, the available material spans more than one year of festivities, indicating participants likely gathered flowers from multiple locations over three years. The findings presented are specific to the study period, and we caution that they may vary annually depending on where extraction occurs.

Morphometric methods have been successfully employed to determine the unknown origins of organisms in various animal groups, utilizing live specimens, preserved in museums or commercialized [26,27,28,29,30]. However, they are relatively unexplored in plant species for this purpose. This study represents an initial attempt to infer the origins of Zaachila individuals based on morphological data. We point out the challenges (even with other types of data, e.g., genetics), including potential variance due to collection from multiple localities during the sampling period, gaps in samples from other distribution sites [26,28], and varying population sizes [40], which are intrinsic to orchids and other epiphytes [41].

Several populations studied here are situated in areas where specimens are harvested for local markets and religious use [3,8,9], impacting genetic variability and effective population sizes, as well increasing the chances of experiencing inbreeding depression and bottleneck events [13,14,15]. Genetic analysis could help elucidate the causes of floral morphological variation in *P. karwinskii* and its biogeographical patterns, though correlations between genetic and morphological variability are not always straightforward.

Mexico’s use of plants to satisfy aspects of the cultural and social life of local communities is vast, given the high cultural and biological diversity [42,43,44]. Among them, ceremonial uses constitute a cultural element of the people, making their transmission important. However, the extraction of wild plants used in such ceremonies often has a negative impact on their populations [45,46]. The conservation and use of wildlife are controversial [47], but these topics must be addressed through collective strategies [45,48]. There are some preliminary initiatives for *P. karwinskii* [10,49]. These efforts motivate us to develop collaborative conservation strategies that ensure the maintenance of the morphological and genetic variability of the species in regions most susceptible to flower extraction for traditional uses, as well as in the communities where these extracted specimens are destined. This will help guarantee the long-term preservation of both the species and its cultural significance.

## 4. Materials and Methods

### 4.1. Biological Material

During the flowering season of *P. karwinskii* (March–April), localities (populations) representing its distribution in Oaxaca were visited between 2015 and 2021. The geographical information of the sampled localities is presented in Table 3, while Figure 5 shows their geographical distribution on a map of Oaxaca. In each visited locality, one flower per individual was collected, ensuring they were on different host trees to avoid collecting ramets from the same individual; the proximal flower on the inflorescence was chosen when it had more than one. The sample size of each locality depended on the population abundance and is indicated in Table 3. A voucher specimen (herbarium or spirit) was prepared from each locality, which was deposited in the Herbarium OAX (acronym according to [50]. Additionally, flowers rescued from specimens used as decorations in Catholic temples during the Easter celebration in the Villa de Zaachila, Oaxaca (2017–2019) were obtained. These samples, whose original locality is unknown, were obtained with permission from the organizing committee of this commemoration in the community in 2019, once they were removed from the temples. Both the flowers collected in the field and the rescued ones were preserved in a fixing solution of water (78%), ethanol 96% (21%), lactic acid 85% (6%), benzoic acid (0.5% *w*/*v*), and glycerin (5%), and then deposited in the Laboratory of Extraction and Analysis of Natural Products (CIIDIR Oaxaca, Instituto Politecnico Nacional).

### 4.2. Selection of Floral Characters

Each flower of *P. karwinskii* was dissected into sepals, petals, labellum, and column to select the characters used as variables in the morphometric analyses, according to Borba et al. [18] and da Cruz et al. [23] with some modifications. Photographs (Canon Rebel camera) were taken of the sepals, petals, and labellum, which were spread out as shown in Figure 6A. The column was separated from the rest of the perianth for recording the characters on this structure, as shown in Figure 6B. The ovary and column were longitudinally sectioned to show the nectary or cuniculus and to record their characters, as shown in Figure 6C. Since the flower of *P. karwinskii* has bilateral symmetry, only the right-side sepal, petal, lateral lobe of the labellum, and lateral tooth of the column were considered. A total of 45 characters were selected from these structures, of which 39 were linear measurements and 6 were angles (Table 4, Figure 6). Each linear measurement was taken with a digital caliper, and the angles were measured with a protractor. Missing data in some individuals for certain characters, due to herbivory, were replaced with the population average for that variable. The set of these continuous characters was recorded in a total of 185 individuals of *P. karwinskii*. Specimens rescued from Zaachila were considered as a separate population. Specimens morphologically similar from populations very close to each other (<5 km apart) were integrated as a single population (expanded population); when a locality was represented by fewer than five specimens, they were also integrated with those from the nearest locality, thus avoiding the effect of a non-representative population. Due to this, the CVA and cluster analysis included 12 expanded populations, as shown in Table 3.

### 4.3. Statistial Analyses

An analysis of variance (ANOVA) and a Tukey test as a post-hoc analysis were conducted to find significant differences among each of the 45 floral characters and the origin locality of the individuals. Out of the 45 floral traits, 5 did not meet all the assumptions for applying an ANOVA, including normality. To assess if there are differences between populations for these five characters (angle at the apex of the middle lobe, thickness at the level of the anther, thickness in the middle part, width at the stigma level, and width of the middle tooth), a Kruskal–Wallis non-parametric test was applied, followed by Dunn’s test with Bonferroni correction as a post-hoc analysis. These analyses were performed using R 2023.12.0 [51] in the stats package.

Multivariate analyses were implemented in Statistica 10 [52]. First, a correlation test was conducted between all possible pairs of variables. Out of the 45 selected variables, 12 showed a correlation greater than 0.9 with each other and were therefore discarded for further analyses. By floral structure, the discarded variables were from the lateral sepal (width at 1/3 of its length, width at 2/3 of its length), dorsal sepal (total length, maximum width, width at 1/3 of its length, width at 2/3 of its length), petal (total length, width at 1/3 of its length, width at 2/3 of its length), labellum (width at 1/3 of its length, width of the base of the middle lobe), and column (thickness in the middle part). The remaining 33 characters were subjected to ordination analysis to evaluate morphological variation among populations, which allowed identifying the most significant variables for morphological patterns. To explore the data structure and detect possible outliers, a principal component analysis (PCA) based on the correlation matrix (correlated variables excluded) was performed. Since this exploratory analysis did not show the presence of outliers, no individuals were removed from the analyses. Subsequently, a canonical variate analysis (CVA) was performed using the expanded populations as categorical variables. We used the standardized coefficients for canonical variables to identify the most important in the observed patterns [18]. Both the correlation test, PCA, and CVA were conducted including and excluding samples from Zaachila (of unknown origin). We calculated a matrix of squared Mahalanobis distances between the expanded population centroids (including Zaachila), using the expanded populations as the group variable and the 33 floral variables. This matrix was used to perform a cluster analysis using UPGMA (unweighted pair-group method) with arithmetic averages as the clustering algorithm.

## 5. Conclusions

Through morphometric analysis, we examined the intraspecific variation in the floral morphology of *P. karwinskii*, identifying characters as potential taxonomic markers for the species related to the *P. citrina* complex. Traits from lateral sepal, labellum, and column, are useful for this purpose and they are related to the morphological patterns of *P. karwinskii*. The majority of floral characters analyzed here varied significantly among populations of *P. karwinskii*. The CVA was informative for discriminating the infraspecific variation in this orchid, but the PCA was not. Albarradas harbors the most differentiated population of *P. karwinskii* in Oaxaca, and it might be recognized as a variety or geographical form of the species and thus considered a priority for conservation. The Teposcolula population has value for horticultural management due to its individuals having the largest flowers in this species.

The results show that the floral characters have the potential to associate specimens of unknown origin with their probable geographical region. The individuals rescued in the Zaachila group with those from Etla and Yanhuitlan suggest that the origin of the former might be assigned to these two localities or others very close to them. Additional studies are recommended to compare morphological variation with genetic and geographical variation of the populations studied here. Finally, this study constitutes an initial attempt to determine the unknown geographical origin of an orchid extracted for religious use by Mexican communities. The method employed here is low-cost, allows the analysis of a large number of individuals, and could be applied to other orchid species. However, the results must be interpreted with caution, as the accuracy of traceability depends on a good reference and a careful analysis.

## Figures and Tables

**Figure 1 plants-13-01984-f001:**
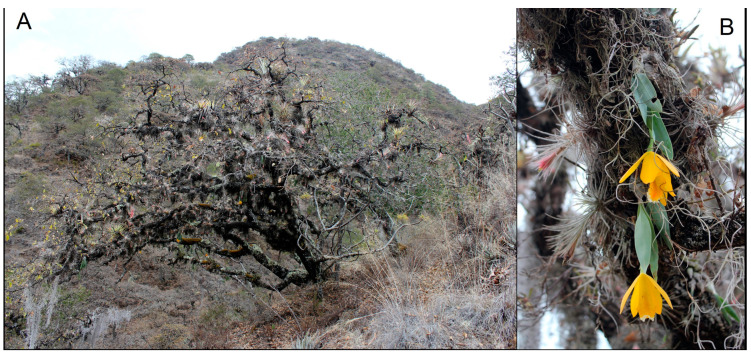
(**A**) Habitat of *Prosthechea karwinskii*, an oak forest in San Pedro y San Pablo Teposcolula, Oaxaca. (**B**) *Prosthechea karwinskii* growing in situ as a hanging epiphyte on *Quercus* sp. in Santo Domingo Yanhuitlan, Oaxaca. Photographs by R. Solano.

**Figure 2 plants-13-01984-f002:**
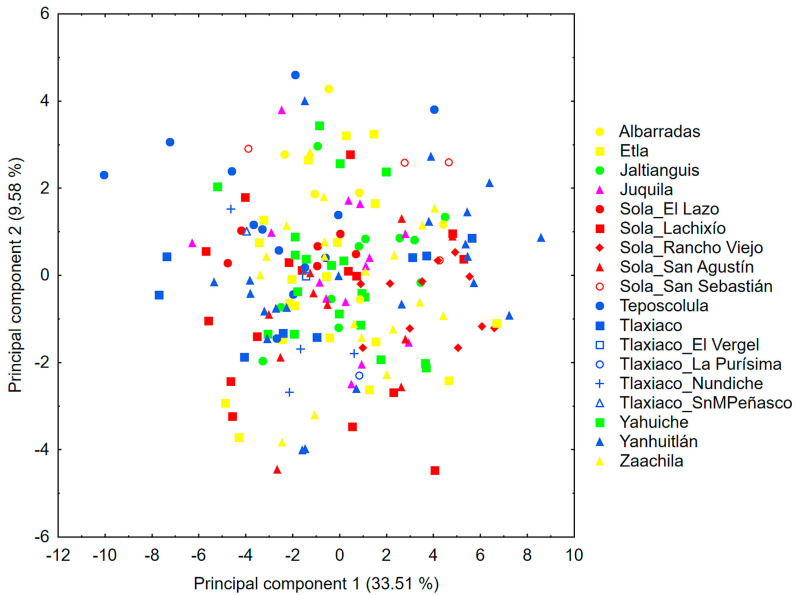
Representation of axes 1 and 2 of the PCA resulting from the variation of 33 floral variables in 185 individuals of *Prosthechea karwinskii* from 18 localities (including Zaachila) in Oaxaca, Mexico.

**Figure 3 plants-13-01984-f003:**
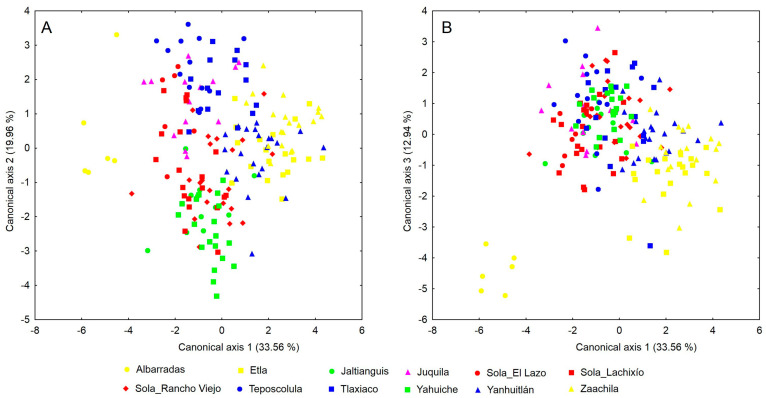
Representation of the axes 1–2 (**A**) and 1–3 (**B**) of the CVA resulting from the variation of 33 floral variables in 185 individuals of *Prosthechea karwinskii* from 12 expanded populations in Oaxaca, Mexico. See Section 4 to see how the expanded populations were integrated.

**Figure 4 plants-13-01984-f004:**
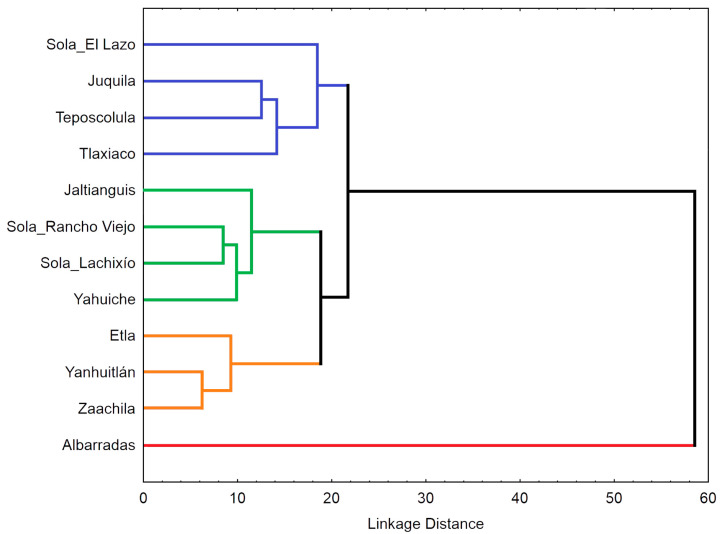
Phenogram showing the relationships between 12 expanded populations of *Prosthechea karwinskii* from Oaxaca, Mexico. The dendrogram was constructed using the UPGMA clustering algorithm and the squared Mahalanobis distances between the population centroids, calculated from 33 floral variables.

**Figure 5 plants-13-01984-f005:**
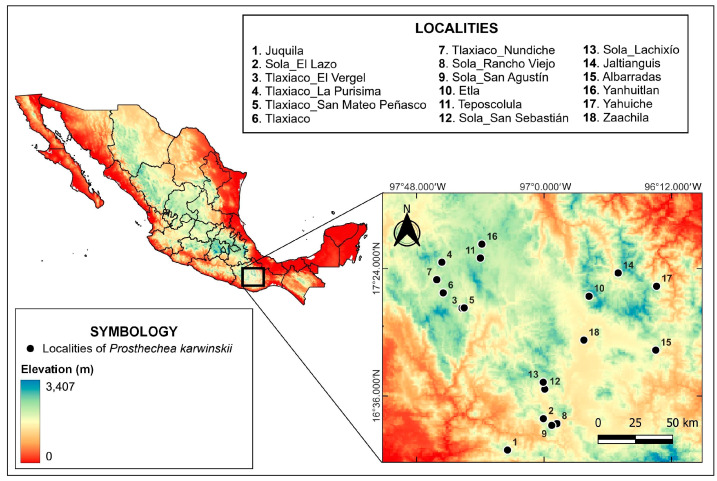
Localities where individuals of *Prosthechea karwinskii* were sampled in Oaxaca, Mexico. See Table 3 for additional information of the localities.

**Figure 6 plants-13-01984-f006:**
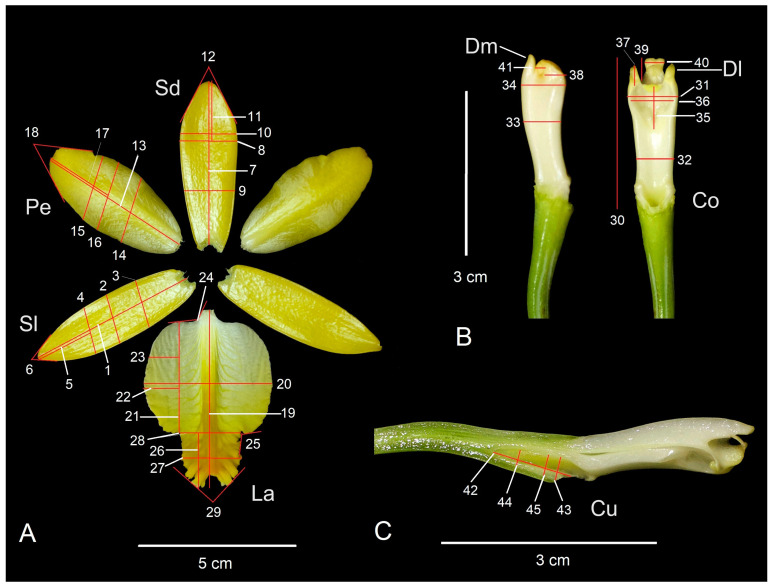
Floral structures showing the variables evaluated in the morphometric analyses of the 18 populations (including Zaachila) of *Prosthechea karwinskii* from Oaxaca, Mexico. (**A**) Flower dissection. (**B**) Lateral and ventral views of the column (**C**) Longitudinal section of the column and ovary, showing the cuniculus. Sl = lateral sepal, Sd = dorsal sepal, Pe = petal, La = labellum, Co = column, Dl = lateral tooth of the column, Dm = median tooth of the column, Cu = cuniculus. See Table 4 for the names of the variables. Photographs by R. Solano.

**Table 1 plants-13-01984-t001:** Results of the analysis of variance (ANOVA) for 40 of the 45 floral characters that met the assumption of normality recorded in 185 individuals from 12 expanded populations of *Prosthechea karwinskii*, including Zaachila. df = degrees of freedom, ss = sum of squares, ms = mean square. Differences are significant at *p* ≤ 0.05, number of asterisks indicates increasing levels of significance, a dot indicates that the test value was not significant.

Variable	df	ss	ms	F-Value	*p*-Value	Significance
SlLt	11	1546	140.51	3.214	<0.001	***
SlAm	11	120.4	10.944	2.213	0.015	*
SlA1	11	127.1	11.552	2.096	0.022	*
SlA2	11	135.1	12.278	2.209	0.022	*
SlLa	11	463	42.13	1.35	0.201	.
SlAa	11	1846	167.86	1.702	0.076	.
SdLt	11	2005	182.27	4.338	<0.001	***
SdAm	11	142	12.913	2.262	0.013	*
SdA1	11	121.9	11.084	1.72	0.072	.
SdA2	11	137.1	12.462	2.177	0.017	*
SdLa	11	467	42.47	1.718	0.072	.
SdAa	11	2178	198	1.513	0.130	.
PeLt	11	1777	161.52	4.154	<0.001	***
PeAm	11	516.8	46.98	3.065	<0.001	***
PeA1	11	473.5	43.05	3.222	<0.001	***
PeA2	11	351.4	31.95	2.592	0.004	**
PeLa	11	515.5	46.86	2.643	0.003	**
PeAa	11	8608	862.5	4.436	<0.001	***
LaLt	11	2123	193	4.351	<0.001	***
LaAm	11	677	61.55	2.497	0.006	**
LaLl	11	678	61.63	2.99	0.001	**
LaAml	11	3.817	0.347	4.113	<0.001	***
LaA1l	11	172.9	15.714	3.881	<0.001	***
LaAul	11	14463	1314.8	9.014	<0.001	***
LaAlm	11	3.817	0.347	4.113	<0.001	***
LaLm	11	5.668	0.5152	4.874	<0.001	***
LaAbm	11	172.6	15.695	1.839	0.050	.
LaAmm	11	169.7	15.423	1.816	0.054	.
CoLt	11	262.2	23.836	6.174	<0.001	***
CoA1	11	46.92	4.266	2.917	0.001	**
RoAl	11	48.02	4.366	3.471	<0.001	***
RoAn	11	42.36	3.85	2.119	0.016	*
DlAl	11	95.07	8.642	14.04	<0.001	***
DlAn	11	27.24	2.4768	3.08	0.008	**
DmAl	11	109.69	9.972	18.58	<0.001	***
AnDlDm	11	12.45	1.1315	6.602	<0.001	***
CuLt	11	112.9	10.264	2.263	0.013	*
CuAm	11	12.1	1.1	3.215	<0.001	***
CuA1	11	1.27	0.11542	3.472	<0.001	***
CuA2	11	7.75	0.7046	2.245	0.014	*

**Table 2 plants-13-01984-t002:** Contribution between variables and axes with eigenvalues > 1.00 of PCA (18 populations, including Zaachila) and CVA (12 expanded populations) of *Prosthechea karwinskii* from Oaxaca, Mexico. Values in bold correspond to the three highest for each axis (A).

Variable	PCA (*n* =185)								CVA (*n* =185)		
	A1	A2	A3	A4	A5	A6	A7	A8	A1	A2	A3
SlLt	−0.835	0.367	0.015	0.063	0.114	0.132	0.030	0.031	−0.128	**−0.941**	−0.066
SlAm	**−0.852**	−0.089	−0.055	0.070	−0.247	0.085	0.067	−0.019	**−0.551**	0.230	−0.146
SlLa	−0.671	0.369	−0.209	0.087	−0.050	0.187	−0.059	−0.117	0.435	−0.159	−0.004
SlAa	−0.281	**−0.701**	0.027	0.372	−0.134	0.054	−0.001	−0.050	0.411	0.241	0.222
SdLa	−0.657	0.403	−0.166	0.028	−0.041	0.161	−0.051	−0.242	−0.230	0.006	0.112
SdAa	−0.174	**−0.696**	−0.001	0.407	−0.121	−0.033	0.056	0.161	−0.242	−0.121	−0.024
PeAm	**−0.853**	0.104	−0.045	0.114	−0.221	0.025	0.059	0.092	0.062	0.352	−0.007
PeLa	−0.706	0.407	−0.128	0.124	0.002	0.043	0.032	−0.164	0.082	−0.188	0.205
PeAa	−0.176	**−0.632**	0.114	**0.457**	0.059	0.189	0.016	0.129	0.341	−0.104	0.191
LaLt	−0.827	0.323	0.106	0.103	0.121	0.063	−0.067	0.153	0.158	**1.015**	0.088
LaAm	**−0.861**	0.069	0.069	0.236	−0.244	−0.081	−0.007	0.081	0.291	−0.420	**0.480**
LaLl	−0.804	0.201	0.147	0.205	−0.006	−0.021	−0.040	0.150	0.143	0.241	−0.436
LaAml	−0.630	0.199	0.145	0.204	−0.376	−0.276	−0.052	0.184	**−0.533**	0.040	−0.050
LaAul	0.251	−0.043	**−0.518**	−0.069	0.206	0.137	0.018	−0.242	0.082	−0.230	**0.628**
LaAlm	−0.074	0.020	−0.022	−0.010	**0.474**	**0.441**	**0.375**	**0.404**	0.034	−0.040	0.157
LaLm	−0.604	0.329	0.096	0.049	0.147	0.192	−0.094	−0.031	0.227	−0.273	−0.064
LaAmm	−0.661	−0.047	−0.018	0.131	−0.121	0.270	0.028	−0.166	−0.432	−0.050	−0.139
LaAam	0.016	−0.220	0.070	0.026	−0.102	**0.458**	**−0.717**	−0.011	0.049	−0.104	−0.053
CoLt	−0.622	0.168	−0.107	0.016	0.249	−0.313	−0.037	0.161	−0.152	0.557	−0.050
CoAe	−0.729	−0.390	−0.297	−0.277	−0.039	−0.125	0.024	0.078	−0.497	−0.334	0.034
CoA1	−0.711	−0.311	−0.327	−0.208	−0.065	0.094	0.204	−0.065	0.031	−0.074	−0.076
CoGa	−0.750	−0.274	−0.345	−0.195	−0.002	−0.090	0.233	−0.026	0.081	0.142	**0.484**
RoAl	−0.394	−0.147	−0.248	**−0.405**	0.094	−0.156	**−0.488**	0.207	−0.327	−0.198	0.354
RoAn	−0.588	−0.372	−0.350	−0.404	0.009	−0.062	−0.121	0.145	0.285	0.219	−0.269
DlAl	−0.339	0.008	0.067	0.189	**0.580**	−0.345	−0.163	0.260	0.162	**0.594**	0.339
DlAn	−0.543	−0.199	−0.204	0.021	0.137	−0.316	0.028	**−0.297**	−0.240	−0.393	0.309
DmAl	−0.252	−0.253	0.140	0.261	**0.505**	0.237	−0.057	−0.193	**0.950**	−0.243	−0.160
DmAn	−0.494	−0.329	−0.096	−0.040	0.262	0.044	−0.071	−0.283	0.108	−0.213	−0.342
AnDlDm	−0.089	−0.043	0.019	**−0.489**	−0.278	**0.463**	0.081	**0.314**	−0.311	−0.402	−0.360
CuLt	−0.466	−0.051	0.303	−0.273	0.306	0.073	−0.035	−0.019	−0.098	0.211	−0.088
CuAm	−0.424	−0.186	**0.651**	−0.201	−0.088	−0.131	−0.102	−0.133	0.144	0.206	−0.107
CuA1	−0.486	−0.126	**0.647**	−0.315	0.108	0.013	0.117	−0.088	0.217	0.221	−0.202
CuA2	−0.484	−0.218	0.622	−0.356	−0.026	−0.042	0.146	−0.169	−0.096	−0.088	0.016
Eigenvalue	11.06	3.16	2.35	1.93	1.65	1.42	1.13	1.03	3.29	1.96	1.27
Accumulated variance (%)	33.51	43.10	50.23	56.09	61.08	65.37	68.80	71.91	33.56	53.52	66.46

**Table 3 plants-13-01984-t003:** Geographic information for the localities of *Prosthechea karwinskii* in Oaxaca, Mexico. See Section 4.2 to know how the expanded populations were integrated. QF = *Quercus* forest, QPF = *Quercus*-*Pinus* forest. *n* = number of individuals represented in the sample size per population. The locality/population numbers correspond to those shown in the map of Figure 5. The species is an orchid protected by Mexican environmental legislation; therefore, the coordinates of the localities were omitted. NA = the data does not apply.

Number	Locality/Population	*n*	Expanded Population (*n*)	Elevation	Vegetation
1	Amialtepec, Santa Catarina Juquila	14	Juquila (14)	2105 m	QF
2	El Lazo, Sola de Vega	6	Sola_El Lazo (6)	1840 m	QPF
3	El Vergel, Tlaxiaco	1	Tlaxiaco (15)	1900 m	QPF
4	La Purisima, Tlaxiaco	1		2160 m	QPF
5	San Mateo Peñasco, Tlaxiaco	1		2150 m	QF
6	Tlaxiaco	8		2160 m	QPF
7	Santiago Nundiche, Tlaxiaco	4		2200 m	QPF
8	Rancho Viejo, Sola de Vega	11	Sola_Rancho Viejo (21)	1838 m	QF
9	San Agustin, Sola de Vega	10		2091 m	QPF
10	San Agustin, Etla	19	Etla (19)	1950 m	QPF
11	San Pedro y San Pablo Teposcolula	13	Teposcolula (13)	2410 m	QPF
12	El Vado-San Sebastian de las Grutas, Sola de Vega	4	Sola_Lachixio (20)	2100 m	QF
13	San Vicente Lachixio, Sola de Vega	16		2240 m	QPF
14	Santa Maria Jaltianguis	11	Jaltianguis (11)	2164 m	QPF
15	San Lorenzo Albarradas	6	Albarradas (6)	2230 m	QPF
16	Santo Domingo Yanhuitlan	22	Yanhuitlan (22)	2400 m	QPF
17	Yahuiche, Ixtlan de Juarez	19	Yahuiche (19)	2017 m	QF
18	Zaachila	19	Zaachila (19)	NA	NA

**Table 4 plants-13-01984-t004:** Floral morphologic variables of *Prosthechea karwinskii* and their coding used in the present study. The number and code of these variables correspond to those shown in Table 1 and Table 2, as well as in Figure 6.

Floral Structure	Variable	Number	Code
Lateral sepal	Total length	1	SlLt
	Maximum width	2	SlAm
	Width at 1/3	3	SlA1
	Width at 2/3	4	SlA2
	Length between maximum width and apex	5	SlLa
	Angle at the apex	6	SlAa
Dorsal sepal	Total length	7	SdLt
	Maximum width	8	SdAm
	Width at 1/3	9	SdA1
	Width at 2/3	10	SdA2
	Length between maximum width and apex	11	SdLa
	Angle at the apex	12	SdAa
Petal	Total length	13	PeLt
	Maximum width	14	PeAm
	Width at 1/3	11	PeA1
	Width at 2/3	16	PeA2
	Length between maximum width and apex	17	PeLa
	Angle at the apex	18	PeAa
Labellum	Total length	19	LaLt
	Maximum width	20	LaAm
	Length of the lateral lobe	21	LaLl
	maximum width of the lateral lobe	22	LaAml
	Width at 1/3 of lateral lobe	23	LaA1l
	Angle between lateral lobe and claw	24	LaAul
	Angle between lateral and middle lobes	25	LaAlm
	Length of middle lobe	26	LaLm
	Maximum width of middle lobe	27	LaAmm
	Width at the base of the middle lobe	28	LaAbm
	Angle at the apex of the middle lobe	29	LaAam
Column	Total length	30	CoLt
	Width at the stigma level	31	CoAe
	Width at 1/3	32	CoA1
	Thickness in the middle part	33	CoGm
	Thickness at the level of the anther	34	CoGa
Rostellum	Height of the stigmatic cavity	35	RoAl
	Width of the stigmatic cavity	36	RoAn
Teeth of the column	Height of the lateral tooth	37	DlAl
	Width of the lateral tooth	38	DlAn
	Height of the middle tooth	39	DmAl
	Width of the middle tooth	40	DmAn
	Width of the gap between the middle and the lateral teeth	41	AnDlDm
Cuniculus	Total length	42	CuLt
	Maximum width	43	CuAm
	Width at 1/3	44	CuA1
	Width at 2/3	45	CuA2

## Data Availability

The authors assure that, by mutual agreement, we have shared data and information included in this study for its potential publication. We note that access to the Appendix A accompanying the manuscript should be restricted until publication is finalized.

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
