# Peer review of "Variation in the Floral Morphology of Prosthechea karwinskii (Orchidaceae), a Mexican Endemic Orchid at Risk"

_plants, 2024, doi:10.3390/plants13141984_

Round 1

Reviewer 1 Report

Comments and Suggestions for Authors

The paper contains a large amount of data, which makes it worth publishing.  It is not a groundbreaking paper and the importance of the results is restricted to the species Prosthechea karwinskii studied here, which makes it less attractive to a broad audience.

The text is clearly written and easily understandable, but would deserve editing by a native English speaker - although the language is not bad, some improvements could still be made. The statistical analysis is adequate.

Specific comments:

Table S1 should clearly state what is what: localities in columns, floral characteristics in rows - and the meaning of the abbreviations of the letter must be explained by the reference to the main text. The meaning of "medidas" is unclear - an unknown word. The meaning of "no significant differences" is unclear. It must be clearly declared that these are comparisons in rows and that the same letters mean the differences within rows are insignificant. Reference to Table 4 should be given for the meaning of the variables.

Comments on the Quality of English Language

No comments

Author Response

For research article

Response to Reviewer 1 Comments

1. Summary

Thank you very much for taking the time to review this manuscript. Please find the detailed responses below and all the corresponding revisions/corrections highlighted/in track changes in the re-submitted files.

2. Questions for General Evaluation

Reviewer’s Evaluation

Response and Revisions

Does the introduction provide sufficient background and include all relevant references?

Yes

NA

Is the research design appropriate?

Yes

NA

Are the methods adequately described?

Yes

NA

Are the results clearly presented?

Yes

NA

Are the conclusions supported by the results?

Yes

NA

3. Point-by-point response to Comments and Suggestions for Authors

Comments 1: The paper contains a large amount of data, which makes it worth publishing. It is not a groundbreaking paper and the importance of the results is restricted to the species Prosthechea karwinskii studied here, which makes it less attractive to a broad audience.]

Response 1: Thank you for pointing this out. Although our study is restricted to a species of epiphytic orchid, it concerns a taxon that has long been included within another morphologically similar species, P. citrina, which has a very broad delimitation, and we now know that it includes several very similar species. Our study focused on the best-known species of the group, and the method used here has the potential to define the taxonomic boundaries of the remaining members, as well as to identify unique forms of interest for conservation and management. This argument is included in the discussion and conclusions sections of the manuscript, in text highlighted in yellow color.

Comments 2: Table S1 should clearly state what is what: localities in columns, floral characteristics in rows - and the meaning of the abbreviations of the letter must be explained by the reference to the main text. The meaning of "medidas" is unclear - an unknown word. The meaning of "no significant differences" is unclear. It must be clearly declared that these are comparisons in rows and that the same letters mean the differences within rows are insignificant. Reference to Table 4 should be given for the meaning of the variables.

Response 2: Agree. We have, accordingly, revised and modified the text of the legend for Table S1 to emphasize this point. We have changed the Spanish word (medidas) to the correct English term (floral variable); we have explained in the legend what constitutes a significant difference; we have indicated that the table includes average values compared between localities, that the letters following these values indicate whether they are statistically similar or different. Finally, we refer to Table 4 so the reader can consult the names of the variables included in Table S1.

4. Response to Comments on the Quality of English Language

Point 1: The text is clearly written and easily understandable, but would deserve editing by a native English speaker - although the language is not bad, some improvements could still be made. The statistical analysis is adequate.

Response 1: Agree. We have, accordingly, revised and modified the manuscript text in order to attend this observation.

5. Additional clarifications

1.     All changes made to the manuscript text compared to the original version submitted to the journal are highlighted in yellow for easy reference.

2.     The text in Results and Discussion sections was modified to make it more understandable by including the names of the variables instead of their abbreviations.

3.     The Figure 3 was modified because the labels for each of the graphs were missing.

4.     We made a couple of improvements in Figure 5 (map). Specifically: 1) we added the localities directly in the figure, for easier identification of the sampled localities, 2) in the general map, we corrected the position of the box that shows the enlarged area.

5.     Table 1 was modified to adjust the number of decimal places in the probability values, and a column was included to show the significance level of the statistical test

6.     The conclusions briefly describe how the two predictions from our study were met.

7.     We corrected some writing errors throughout the text, unnoticed in the initial version (insertions in track changes highlighted in yellow).

Reviewer 2 Report

Comments and Suggestions for Authors

The manuscript with the title “Variation in the floral morphology of Prosthechea karwinskii (Orchidaceae), a Mexican endemic orchid at risk” presents a morphologic study of flowers of an epiphyte orchid species that is endemic to Mexico. Given habitat loss and illegal trade, the comparative study on this orchid populations provides valuable insight on its phenotypic diversity and could help conservation programs.

The Abstract provides a very good summary of the manuscript, and highlights the main findings.

Introduction

Provides a good background for the study and 2 objectives for the research.

Results

It is difficult to follow the explanations and presentation of the results because of the use of acronyms; hence I suggest in text to use a more accessible way of presenting the results referring to the categories of observations as presented at material and method. For example, three dorsal sepal variables (SdLt, …) and one labellum variable (LaLt) did not show significant differences among localities. This would greatly help readers to understand, and follow more easily the results. Right now, it is difficult by reading the manuscript to comprehend it.

Page 9 – I suggest that p-values to be presented only until 3 decimals (such as 0.201 …. 0.001) and those lower simply as <0.001. Table shall have a title caption, and perhaps another column could be added in which significance to be inserted: such as *, **, *** by comparing p-values with thresholds of 0.05, 0.01 and 0.001.

Conclusions

At the end of the introduction the authors proposed two objectives.

1) … recognize the inter-population variation …

2) … morphological marker associated with the geographical origin …

The conclusions have to respond to these two objectives, in order to consider the authors reached their aim, and perhaps conclusions shall have two paragraphs as well.

Best regards.

Comments on the Quality of English Language

minor English style improvements are recommended

Author Response

Comments 1: The manuscript with the title “Variation in the floral morphology of Prosthechea karwinskii (Orchidaceae), a Mexican endemic orchid at risk” presents a morphologic study of flowers of an epiphyte orchid species that is endemic to Mexico. Given habitat loss and illegal trade, the comparative study on this orchid populations provides valuable insight on its phenotypic diversity and could help conservation programs.

The Abstract provides a very good summary of the manuscript, and highlights the main findings.

Introduction

Provides a good background for the study and 2 objectives for the research.]

Response 1: The authors appreciate the reviewer’s opinion on our manuscript.

Comments 2: It is difficult to follow the explanations and presentation of the results because of the use of acronyms; hence I suggest in text to use a more accessible way of presenting the results referring to the categories of observations as presented at material and method. For example, three dorsal sepal variables (SdLt, …) and one labellum variable (LaLt) did not show significant differences among localities. This would greatly help readers to understand, and follow more easily the results. Right now, it is difficult by reading the manuscript to comprehend it.

Response 2: We appreciate the suggestion to improve the manuscript. We have modified the text presenting the results by replacing the acronyms of the floral variables with their full names to make it easier for the reader to understand. This change was also applied to the discussion, where the floral variables are mentioned.

Comments 3: [Page 9 – I suggest that p-values to be presented only until 3 decimals (such as 0.201 …. 0.001) and those lower simply as <0.001. Table shall have a title caption, and perhaps another column could be added in which significance to be inserted: such as *, **, *** by comparing p-values with thresholds of 0.05, 0.01 and 0.001.]

Response 3: We appreciate the suggestion to improve the manuscript. We have modified Table 1 to adjust the number of decimals in the probability values; p-values with more than four decimal places were simplified to <0.001; and an additional column was included to show the significance level of the statistical test: 0.1, 0.01, 0.001, <0.000.

Comments 4: [Conclusions

At the end of the introduction the authors proposed two objectives.

1) … recognize the inter-population variation …

2) … morphological marker associated with the geographical origin …

The conclusions have to respond to these two objectives, in order to consider the authors reached their aim, and perhaps conclusions shall have two paragraphs as well.]

Response 4: In conclusions the text was modified to describe how the two objetives from our study were met, this text is highlighted in yellow.

4. Response to Comments on the Quality of English Language

Point 1: [Minor English style improvements are recommended]

Response 4: Agree. We have, accordingly, revised and modified the manuscript text in order to attend this observation.

5. Additional clarifications

  1. The results text was modified to make it more understandable by including the names of the variables instead of their abbreviations.
  2. We made a couple of improvements in Figure 5 (map). Specifically: 1) we added the localities directly in the figure, for easier identification of the sampled localities, 2) in the general map, we corrected the position of the box that shows the enlarged area.
  3. The Figure 3 was modified because the labels for each of the graphs were missing.
  4. We corrected some writing errors throughout the text, unnoticed in the initial version (insertions in track changes highlighted in yellow).
